# Fast and Deep Diagnosis Using Blood-Based ATR-FTIR Spectroscopy for Digestive Tract Cancers

**DOI:** 10.3390/biom12121815

**Published:** 2022-12-05

**Authors:** Shanshan Guo, Gongxiang Wei, Wenqiang Chen, Chengbin Lei, Cong Xu, Yu Guan, Te Ji, Fuli Wang, Huiqiang Liu

**Affiliations:** 1School of Physics and Optoelectronic Engineering, Shandong University of Technology, Zibo 255000, China; 2Department of Clinical Laboratory, Zibo Central Hospital, Zibo 255000, China; 3Shanghai Synchrotron Radiation Facility, Shanghai 201204, China; 4Department of Oncology, Zibo Central Hospital, Zibo 255000, China

**Keywords:** infrared spectroscopy, blood-based molecular biology, infrared molecular fingerprint, machine learning, digestive tract cancers

## Abstract

Attenuated total reflection-Fourier transform infrared spectroscopy (ATR-FTIR) of liquid biofluids enables the probing of biomolecular markers for disease diagnosis, characterized as a time and cost-effective approach. It remains poorly understood for fast and deep diagnosis of digestive tract cancers (DTC) to detect abundant changes and select specific markers in a broad spectrum of molecular species. Here, we present a diagnostic protocol of DTC in which the in-situ blood-based ATR-FTIR spectroscopic data mining pathway was designed for the identification of DTC triages in 252 blood serum samples, divided into the following groups: liver cancer (LC), gastric cancer (GC), colorectal cancer (CC), and their different three stages respectively. The infrared molecular fingerprints (IMFs) of DTC were measured and used to build a 2-dimensional second derivative spectrum (2D-SD-IR) feature dataset for classification, including absorbance and wavenumber shifts of FTIR vibration peaks. By comparison, the Partial Least-Squares Discriminant Analysis (PLS-DA) and backpropagation (BP) neural networks are suitable to differentiate DTCs and pathological stages with a high sensitivity and specificity of 100% and averaged more than 95%. Furthermore, the measured IMF data was mutually validated via clinical blood biochemistry testing, which indicated that the proposed 2D-SD-IR-based machine learning protocol greatly improved DTC classification performance.

## 1. Introduction

Digestive tract cancer (DTC) symptoms are usually non-specific and frequently appear as common symptoms of a benign non-tumor condition, including abdominal pain and diarrhea, fever, and sour eructation [1]. They are often diagnosed as emergency presentations by most general practitioners. Although some are truly linked to a certain early DTC, it’s often difficult to accurately diagnose in general clinics and health check-ups based on routine medical examining techniques. With the extensive or excessive use of endoscopy, CT & MRI, the diagnosis rate of medium and advanced DTCs has significantly increased [2,3]. However, the 5-year survival rate of DTC patients has not distinctly improved over the decades [4,5]. Therefore, the timely diagnosis of a certain early DTC with non-specific symptoms, including liver cancer (LC), gastric cancer (GC), and colorectal cancer (CC), is still challenging and becoming a key strategy for treating cancer. Recent studies have shown that metabolic characteristics of biomacromolecules, such as protein dysregulation, can be considered a diagnostic and prognostic marker for early cancer screening [6,7]. Proteins with tissue or blood cells derived from necrotic and apoptotic processes enter the circulatory system through active secretion or leakage, which offers a potential means of monitoring the early cancer status via probing systemic human biofluids. Tumor metabolism is also reflected in the fluctuations of the complex structure of carbohydrates. Compared with tumor biopsy samples, human serum is easily obtained from peripheral blood with less risk of tumor damage [8,9]. Early cancer screening based on blood biochemistry mainly aims at a single molecular group for a specific pathological condition, such as measuring some given proteins mainly consisting of glycoproteins or small molecule metabolites, which limits its sensitivity and specificity in several respects [10,11,12,13].

The infrared spectroscopy of blood-based biopsies has been increasingly applied to biomedical diagnostics as a time and cost-effective approach. It can provide a molecular vibrational nature of disease-related changes, potentially regarded as infrared molecular fingerprints (IMFs) of a certain disease. The Fourier-transform infrared spectroscopy (FTIR) is widely used in cancer screening, characterized as a simple, label-free, non-invasive technique. Many potential IMFs for disease diagnostics have been reported (e.g., brain [14], lung [15], prostate [16], ovarian [17], and breast [18] cancers; metabolic changes related to health monitoring [19,20]). However, whether IMFs can be systematically used to identify DTC type, subtype and stage remain poorly understood, impeding their applicability for DTC. Human blood composition is influenced by many factors, such as physiological states, age, dietary habits, environmental factors, nutritional status, drug consumption, and genotypic variation [21,22]. The blood testing approach to specific cancer screening must take the reference ranges for bio-macromolecular parameters and adequate sample space into account for determining individual analytes or specific cancer features in a spectral fingerprint [23]. Hence, evaluating the validity of IMFs and tumor spectral markers is essential for a better understanding the range of blood-based biological features across molecular species and for any future application of molecular fingerprinting in cancer screening or disease detection.

Compared to the transmission mode of FTIR measurement, the attenuated total reflection mode of FTIR (ATR-FTIR) is more suitable for investigating the vibrational spectroscopy of biofluid samples due to minimizing interference from water molecules. To obtain the high signal-to-noise ratio of FTIR, the dehydration process of liquid samples is often performed prior to ATR-FTIR experiments, even though it results in the loss of in-situ signals and possibly misleading quantitative analysis [24,25]. Synchrotron radiation (SR) light source has the advantage of brightness at least two or three orders of magnitude higher than traditional light source (globar), with an improved spectral signal-to-noise ratio (SNR) of samples [26]. Thus, SR-based ATR-FTIR microscopy facilitates the high-precision in-situ measurement of fresh samples, especially achieving specific spectrum collections of microregion-of-interest [27]. With the combination of data mining (DM) methods, such as principal component analysis (PCA), Hierarchical cluster analysis (HCA), Partial least squares discriminant analysis (PLS-DA), and machine learning (ML) classifiers, the FTIR-based intelligent diagnosis has been widely employed into Corona Virus Disease 2019 (COVID-19) screening [28], health monitoring [29], disease detection [30], and food security. However, to our knowledge, no attempts have yet been made to investigate the classification, subtype, and stage of different DTCs based on blood-FTIR-based IMFs and DM algorithms.

In this study, the proposed pathway for multi-dimensional refined diagnosis of DTCs, as shown in Figure 1, the different data mining techniques using abundant blood-based ATR-FTIR spectra were combined to reveal multi-dimensional information, helpful for general practitioners to better understand and determine the diagnostic and therapeutic strategies of DTC patients. First, Serum was extracted from a blood sample and adequately used in the measurements of ATR-FTIR microscopy and clinical tumor markers due to serum consisting of important metabolites such as protein, cholesterol, glucose, urea, and triglycerides. They potentially provide more impressive information for both spectroscopic and biochemical tests. Then the SR-based ATR-FTIR microscopy was employed in the IMF collection of DTC serums. Then the significant changes derived from raw and second derivative (SD) FTIR signatures related to different DTCs, were extracted and mutually confirmed by clinical tumor markers, including alpha-fetoprotein (AFP), carcinoembryonic antigen (CEA), carcinoembryonic antigen 125 (CA-125), carcinoembryonic antigen 15-3 (CA-15-3), carcinoembryonic antigen 19-9 (CA-19-9) and carcinoembryonic antigen 724 (CA-724). These key functional groups associated with the connections of potential FTIR and early clinical tumor markers were revealed and facilitated comprehensive DTC diagnosis. Importantly, a novel in-situ FTIR feature set, including significant changes of absorbances and peak positions of lipid, protein, saccharide, and nucleic acid compositions in the range of 900–3500 cm^−1^, was proposed to improve the diagnostic level of DTC, involved in auto-classification, subtypes, and stages. Various multivariate statistical analyses and ML algorithms with our in-situ combined FTIR feature set were quantitatively compared to the traditional FTIR feature set. The optimized DM method of FTIR microscopy without complex sample preparation and testing procedure can be used as a triage tool to provide additional information to inform referral decisions. It’s a step towards exploring a new simple, fast diagnostic model of different DTCs, especially for screening early DTCs with good feasibility in sample preparation and testing.

## 2. Materials and Methods

### 2.1. In-Situ FTIR Measurement

Although the dehydration of samples for FTIR measurements enables the higher spectral SNR, it usually suffers from the deformation of FTIR profiles to some degree and the lack of in-situ information, such as native secondary protein structures, decreases the reliability of the spectral feature dataset. For the collection of in-situ FTIR spectra of DTCs and to avoid major sample preparation steps and artifacts, each liquid blood serum sample of 3 μL was directly measured by using an ATR-FTIR spectrometer rather than the transmission mode of FTIR to better mitigate the absorption effects of water and achieve high SNR and in-situ FTIR spectra of DTCs (Appendix A). It allowed us to accurately evaluate spectral features caused by abundant constituent variations in DTC blood serums, properly related to the screening results of clinical tumor markers for mutual corroboration of data validation (Appendix A). The in-situ aqueous sample was subjected to differential spectroscopy with the dry sample spectrum. The obtained differential spectrum (blue) was compared with the pure water spectrum (red), as shown in Appendix A. It could be observed that the O-H bond in water (3346 cm^−1^) and the H-O-H bond (1637 cm^−1^) combines with the protein, leading to a frequency shift in the difference spectrum: it is assigned to the N-H stretching vibration in the protein amide A band (3371 cm^−1^), protein amide I (1694 and 1613 cm^−1^) and protein amide II (1539 cm^−1^). It can be found that the chemical bonding after water molecule rupture may bind to the organic components in the pairs of original samples, thus affecting the data reliability.

### 2.2. Study Participants

The Ethics Committees from the Central Hospital of Zibo approved the study’s protocols. This study was conducted on 25 LC subjects, 68 GC patients, 73 CC-diagnosed patients, and 44 who did not have a clinical diagnosis of cancer (control healthy individual group). Of note, for patients with DTC, two to three blood specimens from different infection stages before recovery were collected to enrich and generalize the data set. Consequently, 252 blood serum samples were required in this study. The data collected from patients involved in this experiment were subject to the Personal Data Act and Act on Ethical Review for research involving human subjects. All participants were fully informed, and all of them signed consent forms for this study. The details of the participants are depicted in Appendix A.

### 2.3. Blood-Sample Preparation and ATR-FTIR Spectroscopy

The serum samples were obtained by blood centrifugation under 4000 rpm for 10 min. Prior to measurement, the serum samples were stored at −80 °C to ensure viability. The experiments were performed on the BL06B line station at Shanghai Synchrotron Radiation Facility (SSRF, China), and the sera were thawed sufficiently before the experiments. The beamline is equipped with the FTIR infrared spectrometer, microscope and a 64 × 64 FPA detector with ATR attachment. An aliquot of 3μL of the serum sample was transferred onto the ATR crystal for each measurement. Afterward, each spectrum was scanned 32 times cumulatively, and the spectra between 4000 and 600 cm^−1^ were collected subsequently.

Meanwhile, the IR spectra of the dried samples were measured. Water absorption was monitored by OH stretching at around 3300 cm^−1^ and bending at around 1635 cm^−1^. It took about ten minutes for the samples to be sufficiently dried. In addition, transmission mode spectroscopic measurements were performed, where 3 μL of serum was dropped on a barium fluoride (BaF_2_) window with a resolution of 4 cm^−1^, and 64 scans were accumulated for each spectrum. Prior to data analysis, the raw spectral data were preprocessed by OPUS software, and all spectra were preprocessed by linear automatic baseline calibration, smoothing, and normalization. Then, the SD-IR was calculated using the Savitzky-Golay algorithm.

### 2.4. Multivariate and Statistical Analysis

Analysis of variance (ANOVA) [31] is a well-established method to evaluate the difference in the observed statistics among groups by calculating the F-statistic value. The larger F-statistic value yields a lower *p*-value, indicating that the groups are more likely to differ in this statistic. *P* was considered significant at 0.05. In this work, the *p*-value was used to evaluate the statistical spectral differences of major bands in absorbances of DTC patients and healthy controls by the software of Origin 2018. Hence, the potential spectral markers can be primarily identified.

HCA and PCA are widely used unsupervised approaches for analyzing data [32,33]. PCA is a non-parametric method for extracting related information from confusing data sets and provides dimensionality and variable reduction by maintaining variance [34]. In HCA, the Euclidian distance is calculated to construct the hierarchical structure, in which root nodes are formed by spectra with the shortest distances, while leaf nodes are formed by the longest distances. PLS-DA is a supervised chemometric technique whereby the so-called latent variables are successively extracted to find the maximum correlation between the X-matrix and Y-matrix [35,36]. Compared with PCA, PLS-DA is more suitable for identifying the significant wavenumbers for discrimination by inspecting the regression vector or the VIP.

Due to the category imbalance in examining the differences between patients with DTCs and non-cancer patients, the synthetic minority over-sampling technique (SMOTE) sampling method was used throughout the study to ensure that there was no bias in the model [37]. SMOTE is a modification of the random oversampling algorithm. It is a classical method for solving the categorical sample imbalance problem, which finds the k-nearest samples of each sample in the imbalance category by computing the Euclidean distance [38,39]. Subsequently, ML techniques are applied to the spectral datasets [40]. 10-fold cross-validation is used to test the accuracy of the algorithm. The data set is divided into ten parts, nine of which are used as training data and one as test data. The average of the correctness of the ten results is used as an estimate of the accuracy of the algorithm. This approach aims to identify the signals of cancer from a known patient cohort to develop a trained classification model and then use this information to predict the presence of cancer in an unknown population [41,42].

## 3. Results

To investigate whether the serum-based ATR-FTIR spectroscopy is capable of refined diagnosis of DTC, we measured the in-situ FTIR spectroscopic variabilities of liquid blood serum in the range from 900 to 3500 cm^−1^, including spectral peak absorbance and wavenumber-shift, which were quantitatively analyzed to reveal spectral markers and comprehensively used to build the multivariate feature dataset of ML for identification of different DTC types, subtypes or stages.

### 3.1. Spectrally Resolved Band Assignments and Differential Interpretation of DTCs

To minimize the influence of individual differences, the averaged spectra and extracted ATR-FTIR obtained IMFs of four groups of blood serum samples. They showed the representative absorption bands of nucleic acid, carbohydrates, proteins, and lipids related to DTCs (Figure 2a). The SD-IR were calculated to sensitively reveal the variations of vibration-absorbed bands of biomacromolecules dominated by amide bands, including amide A (3365 cm^−1^), amide I (1639 cm^−1^), amide II (1544 cm^−1^) and amide III (1240 cm^−1^) attributed to the vibrations of protein backbone (Figure 2b). Compared to the control group, it’s found that the amide I band of DTC were all significantly redshifted by 2 cm^−1^, indicating variations of protein secondary structures (*p* < 0.001 in Table 1). The amide II band of CC has a significant redshift of 10 cm^−1^, and its absorbance also is lower than others (*p* < 0.001, Table 1). The CH_2_ bending, COO^−^ stretching, and OH bending vibrations centered at 1455 cm^−1^, 1398 cm^−1^, and 1354 cm^−1^ with significant changes related to carboxylate. The peaks at 1309 cm^−1^, 1166 cm^−1^, and 1082 cm^−1^ reflected the CH bending, C-O-C stretching and PO^2−^ stretching vibrations, respectively, linked to the characteristics of nucleic acid. The NH-C=O vibration at 1117 cm^−1^ originated from the carbohydrate ring, which helps understand the glycoprotein variations. Moreover, major molecular vibrations produced a series of peaks around 2965 cm^−1^, 2925 cm^−1^, 2857 cm^−1^, and 1741 cm^−1^, respectively, derived from the symmetric and asymmetric CH_3_, CH_2_, and C=O stretching vibrations. It reflected the characteristic variations of main lipids. Generally, we found that the original spectra had quantitative variations of integrated band locations and relative absorbance between DTC and control groups (*p* < 0.01, Table 1), indicating the significant changes in IMFs of bio-macromolecular conformations. Some spectral peaks were potentially associated with clinical DTC tumor markers.

### 3.2. Comparison of FTIR Spectral and Clinical Tumor Marker Screening

Tumor markers are clinically detected by serum biochemistry due to abundant serum proteins, including human serum albumin (HSA), immunoglobulin (IG), transferrin, and α-antitrypsin. There are three types of test items related to proteins, binding proteins, peptides, and carbohydrates according to their chemical properties listed in Table 2 in detail. CEA was a part of the IG family, the CA series reflected the metabolism of transferrin and carbohydrates, and AFP was attributed to HAS, denoted as a protein-based marker. The connection of test items and SD-IR spectra was mainly determined by the structural formula and main functional groups, and both testing results were mutually corroborated in Table 2. The spectra of function groups exhibited more specificities in both absorbance and wavenumber shift, suggesting the changes in substance concentration and transitions in protein secondary structures. The *δ*(N-H) and *ν*(C-O-H) absorption bands were found at 1508 cm^−1^ and 1040 cm^−1^ in the DTC spectra with average blue-shifts of 1.8 cm^−1^ compared to the control group, and the relative absorptions of LC and GC increased by 4.2% and 4.5%, agreement with the clinical results of CEA test. However, the spectral variations provided more abundant information in the early stage than that of serum biochemistry. The *v_s_*(NH-C=O) and *δ_out_*(N-H) originated from the sugar ring at 1117 cm^−1^, 986 cm^−1^ were derived from the effect of glycosylation, and blue-shifts of 3 cm^−1^ approximately occurred in DTCs. The significant increases were consistent with the excessive results of the series CA test, especially associated with abnormal CA19-9 levels in biochemical tumor markers in LC and GC. Notably, LC produced a red-shift at 1639 cm^−1^, and a significant absorbance increase occurred up to 100% attributed to *v*(C=O) from the vibration of the α-folded structure in HSA. This finding was also associated with the elevated clinic level of AFP, which could be regarded as a spectral tumor marker of LC. In addition, the obvious variations in the SD-IR spectra, including *v_as_*(C-O-C) at 1166 cm^−1^ from phospholipids, triglycerides, and cholesterol esters and *v_as/s_*(CH_2_) at 2925 and 2898 cm^−1^, indicated that the possible conformational changes in the lipid composition were closely related to DTC metabolism.

### 3.3. IMF-Based Identification of DTCs

There were more abundant quantitative variations of fine band locations and relative absorbance in SD-IR spectra between IMFs of DTC and control groups (*p* < 0.001, Appendix A), suggesting definite transitions in protein secondary structures and significant changes in functional groups. It’s usually used to identify disease status by combining these spectral variations into a multivariant absorbance dataset in conventional unsupervised classification approaches, such as PCA and HCA. Both methods were employed in the absorbance dataset of SD-IR spectra to identify different types of DTCs (Appendix A). The results showed that PCA and HCA classified different DTCs, and the HCA performed better than PCA because of the use of IMFs for HCA. For the specificity of the dataset, the SD-IR dataset is better than that of the raw spectral dataset in the range of 900–3300 cm^−1^, and the dataset of the IMF band in the range of 1200–1700 cm^−1^ is superior to that of the full spectral band. To further improve the identification accuracy of DTCs, the supervised classification methods and different datasets of ATR-FTIR were employed and compared to choose the best pathway of diagnostic feasibility. We combined the raw IR data and SD-IR data into the extending multi-dimensional dataset (combined data), which was evaluated by eight different classification methods: back propagation (BP) neural network, K-nearest Neighbor (KNN), random forest (RF), decision tree (DT), and logistic regression, support vector machine (SVM), multiple linear regression (MVLR), and PLS-DA. Each dataset was subsequently downscaled by PCA; 70% training and 30% test sets were created during each iteration, and the prediction performances were estimated by applying 10-fold cross-validation. The comparison of classification accuracy showed that the PLS-DA model based on the combined data performed with an accuracy of 100%, and the performance of SD-IR data was overall better than that of raw IR data due to the more abundant spectral variations in SD-IR data (Table 3).

Hence, we adopted the PLS-DA method to identify different DTC types with all serum samples of DTCs (Figure 3). The latent variables (LV) of PLS-DA with cross-validation were down-dimensioned by LV scores into three principled variables. Then R^2^Y and Q^2^ values were used to evaluate the fitting effect of the PLS-DA model. R^2^Y denotes the percentage of Y-matrix information that the PLS-DA classification model can explain, and Q^2^ is to evaluate the predictive ability of the PLS-DA model. They were evaluated by R^2^Y values (0.991, 0.977, and 0.997) and Q^2^ values (0.775, 0.752, and 0.781), respectively. Generally, R^2^Y close to 1 and Q2 > 0.5 is recognized as well-predictive model predictability. The regression vectors of each selected variable (LV loadings) and the importance of the corresponding variables were visualized in pseudo-color based on the VIP in Figure 3a. The VIP scores were shown in Appendix A and evaluated by the prediction error rates using three cross-validation methods, all exhibiting good predictability of this model (Appendix A). Subsequently, it indicated that VIP > 1 preferred to assess the impact of every single variable in the model, and then the most discriminating bands of LC located at 1683, 1640, 1514, 1086, and 1026 cm^−1^. Similarly, the most discriminative frequency bands of GC are located at 1686, 1655, 1563, 1515, 1356, and 1202 cm^−1^. The band of CC is located at 1773, 1742, 1714, 1470, 1062, and 1038 cm^−1^. This finding is probably consistent with the sub-peaks correspondence in the SD-IR spectrum of LC, GC, and CC, as shown in Appendix A. The area values under the curve (AUC) were calculated with 0.9868, 0.9712, and 0.9853, respectively, shown in Figure 3b. Therefore, the PLS-DA model combined with ATR-FTIR data could yield good sensitivity and specificity when the appropriate AUC threshold was selected as AUC > 95% in this study (CI: 0.9703~0.997).

To further optimize the identification protocol of DTC and further explore the relationship between subtle changes in serum ATR-FTIR spectra and the detection of early diagnostic markers of DTC, multivariate analysis was performed on four spectral ranges of second-order leads. We carried out the PLS-DA classification with combined data respectively in four spectral ranges, including 2800–3300 cm^−1^ (mainly caused by lipid hydrocarbon chains), 1600–1700 cm^−1^ (protein amide I bands), 1400–1500 cm^−1^ (associated with immunoglobulin IgG) and 1200–1400 cm^−1^ (from lipids, proteins, nucleic acids and other biomolecules, such as the overlapping contribution of glycosaminoglycans, immunoglobulin m). The classification results obtained by the PLS-DA method are shown in Figure 3c–f. It can be found that the accuracies of 1600–1700 cm^−1^ and 1400–1500 cm^−1^ regions achieved up to 100% (Figure 3d,e), benefited from the abundant subtle spectral changes and DTC-related information in the range of IMF. In addition, the classification accuracy reached 91% for the 2800–3300 cm^−1^ region (Figure 3c) and 95% for the 1200–1400 cm^−1^ region (Figure 3f). The results demonstrated that the PLS-DA with combined data of IMFs exhibited the potential feasibility of identifying different DTCs.

### 3.4. Machine Learning for Classification of Different Pathological Stage

The above-presented protocol has great potential in identifying different DTC groups (LC, GC, CC groups) and control groups. However, it’s difficult to obtain satisfactory results for different disease stages of a single DTC group due to their highly similar spectra of lipid, protein, sugar, and nucleic acid components in the range of 900–3500 cm^−1^. Each group of DTC staging was divided into three different stages, including nonspecific symptom, cancer, postoperative, and control, namely the 1–4 class. Information on the specific subgroups is shown in Appendix A. To extend the architecture of the feature dataset (from 1-dimensional (1D) to 2-dimensional (2D)) and introduce more differential spectral information, we noticed that the small spectral changes of DTCs provided by ATR-FTIR spectroscopy, including relative absorbance and wavenumber shift, are capable of building 2D feature dataset of ML to improve the ability of stage identification in a single DTC group. For conventional feature dataset architectures based on the absorbance variations of IR or SD-IR spectra, the absorbance eigenvalues are only used to construct feature datasets, discarding the wavenumber shift features due to IMF distortion caused by sample dehydration. Therefore, the in-situ synchrotron-based ATR-FTIR dataset was employed in building four types of feature sets (IR absorbance, IR absorbance + shift, SD-IR absorbance, SD-IR absorbance + shift) and respectively using eight different ML modes for a group of DTC staging classifications related to DTCs (Appendix A). The accuracy of the 2D feature dataset was greatly improved by nearly 20% compared to the conventional 1D feature dataset of IR or SD-IR (Appendix A, Figure 4). The BP method exhibited good overall performances with respect to the accuracy and stability in all three groups. The in situ 2D-SD-IR-BP-based ML protocol was performed for the application of stage classification (Figure 4). It was found that the accuracy reached above 95.0% for all three groups, superior to conventional 1D-SD-IR-BP-based ML.

## 4. Discussion

Compared to clinical blood biochemistry focusing on the defined analytes, in-situ blood ATR-FTIR measurements can spectral characterize all classes of biomolecular species related to changes in metabolic reaction products and enzyme activities in serum. The IMFs enable the probing of potential spectral markers for early screening of DTCs, as our proposed data mining protocol suggested. Especially, the synchrotron-based ATR-FTIR was applied to native liquid serum samples to quantitatively analyze in-situ spectral features with adequate SNR benefitted from high brilliance IR source. Furthermore, we demonstrated that many IMF markers defined by VIP scores (greater than 1.0) and extracted by R^2^Y and Q^2^ values exhibited good specificity and partially agreed with the blood analytes routinely examined in the clinical laboratory. Our findings lay the foundation for the optimization of disease-specific IMF auto-classifications.

The proposed blood ATR-FTIR data mining protocol exhibited excellent performance for different aspects of diagnosis related to DTCs. It demonstrated that (a) in-situ blood-based ATR-FTIR spectra are essential for building a 2D feature dataset for machine learning, avoiding uselessness of wavenumber shifting distorted by dehydration, (b) IMFs in the range of 1700–1400 cm^−1^ are effectively measurable by ATR-FTIR and provide abundant specific diagnostic information, (c) according to the similarity of DTCs, the PLS-DA and BP ML methods were optimized respectively for inter-group and intra-group classification, (d) SD-IR data is superior to IR data for extraction and utilization of small bio-macromolecular changes (Figure 4 vs. Appendix A). Therefore, the findings are helpful for the exploration of auto-identification feasibility for different levels of DTC diagnosis.

1D-SD-IR-based absorbance dataset is insufficient to distinguish the staging of a certain DTC. Exploring 2D or multi-dimensional feature sets is necessary to improve the classification performance of complex sample systems. For our in-situ serum FTIR data, the quantitative extractions of absorbance and wavenumber shift of vibration peaks guaranteed the construction of the 2D-SD-IR-based dataset for the comprehensive diagnosis of DTCs by setting the reference point of 1000 cm^−1^. Then all changes in band locations can be calculated and combined with the differences in absorbances to form the 2D-SD-IR-based feature dataset. It’s helpful to explore the molecular information at the different staging of DCT via incorporating multivariate analysis. Using the selected DM methods, such as PLS-DA and BP ML classifiers, the identifications of inter-group and intra-group (stages) of DTCs achieved a satisfactory sensitivity and specificity performance.

## 5. Conclusions

In this proof-of-concept study, we presented a methodology of comprehensive DTC diagnosis using in-situ blood-based ATR-FTIR spectroscopy, which optimized a data mining protocol via determination of IMFs, 2D-SD-IR-based feature dataset, classification method, multi-dimensional sample space of DTCs. Taken together, the measured IMFs were linked to the clinical testing results, and the fast and deep spectral diagnosis indicated a step in translating ATR-FTIR into the clinic. There is potential to apply the FTIR-based ML diagnosis to aid the clinical decision as a triage method for DTCs, extending to early cancer screening, health monitoring, disease detection, and food safety. Identifying early DTCs or referral decisions is essential, usually unmet by the current diagnostic pathway.

## Figures and Tables

**Figure 1 biomolecules-12-01815-f001:**
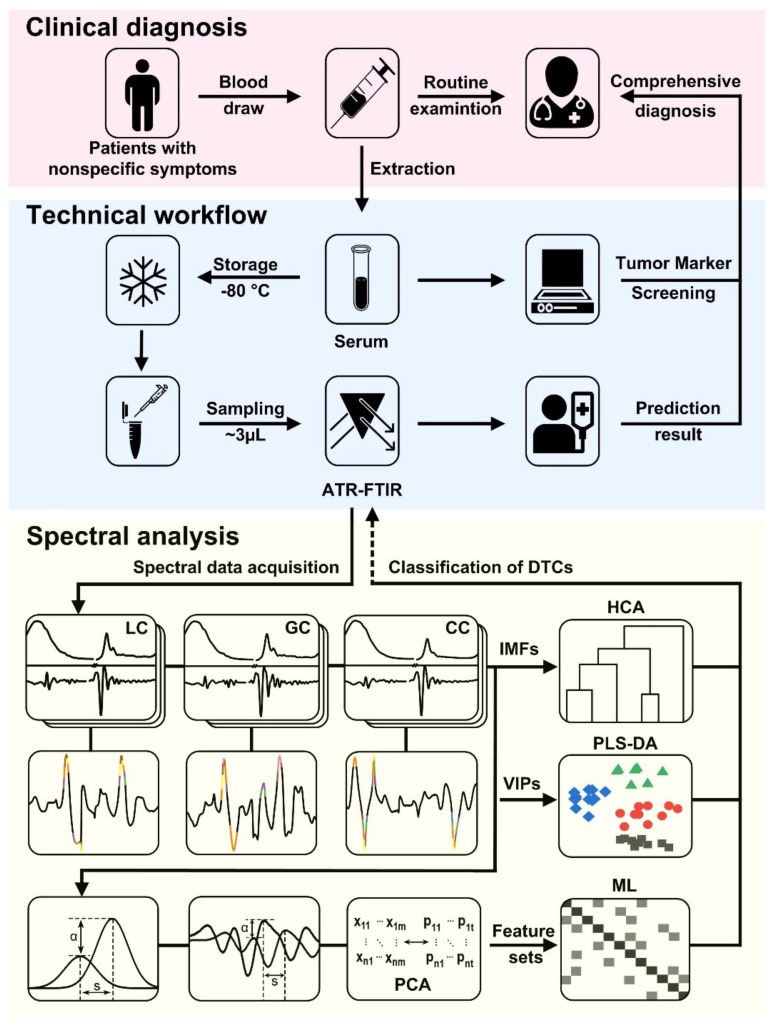
Serum-based ATR-FTIR spectroscopy testing and data mining pathway for accurate DTC triages. The DTC patients with non-specific symptoms could be effectively screened, potentially improving diagnosis efficiency. The different colored symbols (triangles, circles, etc.) represent different groups.

**Figure 2 biomolecules-12-01815-f002:**
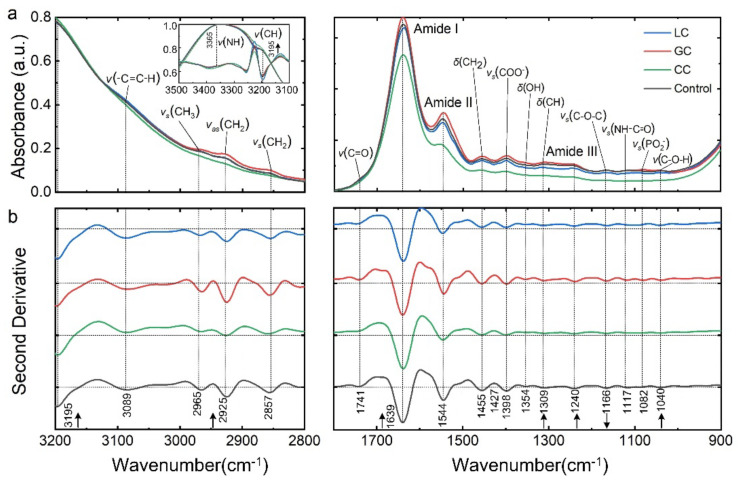
ATR-FTIR profiles of different liquid DTCs, including three types of LC, GC, CC, and control serum samples and its signature analysis of DTCs. (**a**) Average original spectra of DTCs and control samples, annotated with absorption peaks of major molecular vibrations: ν stretching, δ bending, s symmetric, and asymmetric vibrations. (**b**) Average SD-FTIR spectra of DTCs and control samples, characterized by the variations of positions and relative contents of significant molecular compositions.

**Figure 3 biomolecules-12-01815-f003:**
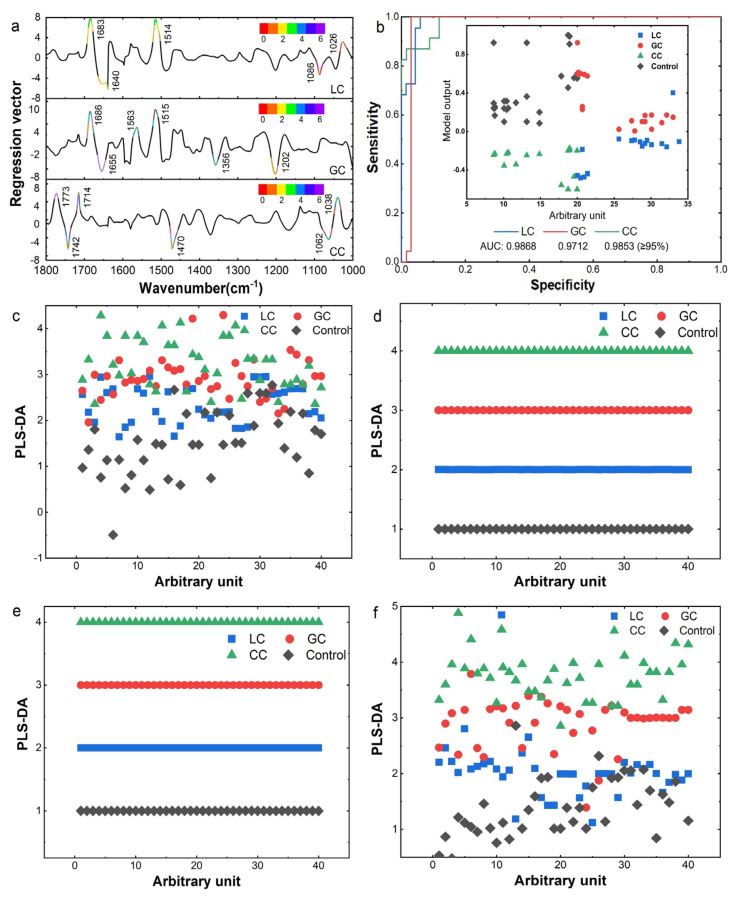
Classification performances of the PLS-DA model for each DTCs vs. controls. (**a**) Regression vectors for LC, GC, and CC classification, respectively. Different ranges of VIP values are shown in different colors. VIP = variable effect on projection. (**b**) ROCs plot of DTC samples (external) versus model output (internal). AUC = area under the curve. (**c**) 3300–2800 cm^−1^ (mainly caused by lipid hydrocarbon chains), (**d**) 1700–1600 cm^−1^ (protein amide I bands), (**e**) 1500–1400 cm^−1^ (associated with immunoglobulin IgG), and (**f**) 1200–1000 cm^−1^ (from lipids, proteins, nucleic acids, and other biomolecules).

**Figure 4 biomolecules-12-01815-f004:**
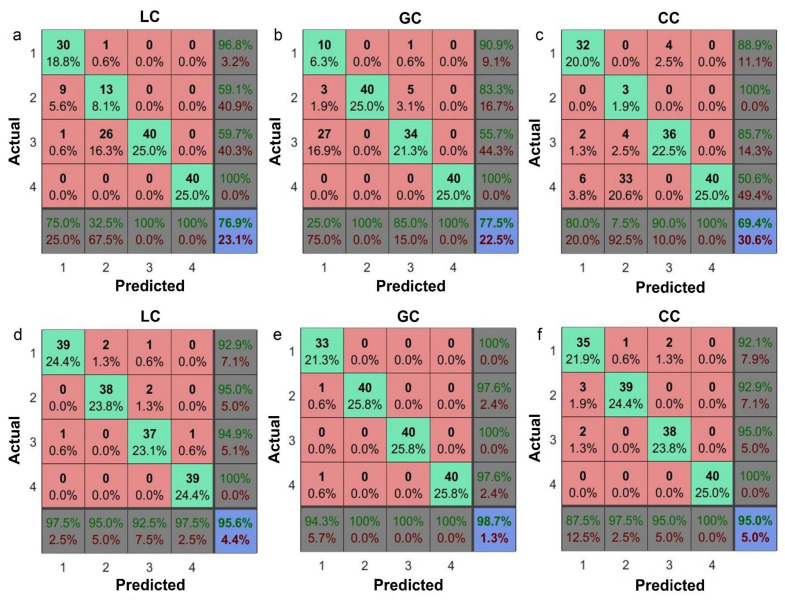
Confusion matrix of the prediction accuracy for SD-IR-based blood of DTC stages. (**a**–**c**) Prediction reports for the SD-IR absorbance feature dataset from patients with DTCs. (**d**–**f**) Prediction results of the optimized SD-IR absorbance + shift feature dataset. The 2D-SD-IR-BP algorithm exhibited the best performance. 1–4: represent detailed staging of patients with DTCs from Appendix A, respectively.

**Table 1 biomolecules-12-01815-t001:** Standardized spectra: Distribution and statistical comparison of DTC and control spectra in terms of band position and relative absorbance.

Assignments	Band Locations/(cm^−1^)				Relative Absorbance (a.u.)			
	LC (*p*)	GC (*p*)	CC (*p*)	Control	LC	GC	CC	Control
amide A	3366.2 (**)	3364.2	3364.2	3346.3	1	1	1	1
*ν*_as_ (CH_3_)	2962.4 (***)	2956.7 (**)	2960.6 (***)	2954.8	1.06	1.12	0.94	1.07
*ν*_as_ (CH_2_)	2927.8 (**)	2925.8	2923.9 (**)	2925.8	1	1.06	0.82	0.83
amide I	1637.5	1639.4 (**)	1638.2 (*)	1637.6	1	1	1	1
amide II	1548.8 (**)	1544.9 (**)	1554.5 (***)	1546.9	0.98	1.04	0.79	1.03
δ (CH_2_)	1456.2 (***)	1455.6 (***)	1460.0 (***)	1452.3	1.11	1.05	0.83	0.83
*ν*_s_ (COO^−^)	1400.0 (***)	1398.3 (**)	1402.2 (***)	1397.6	0.95	1.05	0.73	1.02
amide III	1315.4 (***)	1309.6 (**)	1313.4 (***)	1311.5	0.93	1.26	0.73	1.02
*ν*_as_ (PO^2−^)	1245.9 (**)	1247.9	1244.0 (***)	1247.2	0.87	1	0.67	0.91
*ν*_s_ (C-O-C)	1168.8 (**)	1164.9 (***)	1167.3 (*)	1166.9	0.83	1	0.67	0.91
*ν*_s_ (PO^2−^)	1089.7 (**)	1083.9 (***)	1081.8 (***)	1091.6	0.92	1	0.58	0.91

Liver cancer (LC), gastric cancer (GC), and colorectal cancer (CC). *ν* stretching, *δ* bending, *s* symmetric, *as* asymmetric vibrations. Statistical differences were compared between DTCs and the control group using ANOVA. * *p* < 0.05, ** *p* < 0.01, *** *p* < 0.001.

**Table 2 biomolecules-12-01815-t002:** Comparison and connection of clinic biochemistry and FTIR measurement for serum samples of DTCs.

Serum Biochemistry	StructuralFormula	FunctionGroups	LC	GC	CC
TestItem	ReferenceRange	Results	FTIRBand (Δ)	Absorb(%)	FTIRBand (Δ)	Absorb(%)	FTIRBand (Δ)	Absorb(%)
CEA	0–5ng/mL	LC: 1.37 (±0.07)GC: 34.83 (±0.12)CC: 68.70 (±0.25)	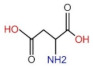	δ(NH)ν(C-O-H)	1508.1↑1041.9↑	0.74↑0.68↑	1508.0↑1039.7	0.72↑0.69↑	1508.1↑1041.2↑	0.73↑0.68
CA	0–35U/mL	LC: 54.74 (±0.21)GC: 201.02 (±0.63)CC: 423.79 (±0.47)	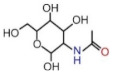	ν*_s_*(NH-C=O)δ*_out_*(NH)	1119.4↑984.8↑	0.68↑0.70↑	1119.4↑986.1↑	0.67↑0.69↑	1118.2↑986.1↑	0.66↑0.70↑
AFP	0–9ng/mL	LC: 701.01 (±0.52)	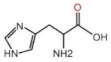	ν(C=O)	1638.2↓	0.02↑				

Liver cancer (LC), gastric cancer (GC), and colorectal cancer (CC). Carcinoembryonic antigen (CEA), carcinoembryonic antigen (CA), alpha-fetoprotein (AFP). Results are in mean (standard deviation); major molecular vibrations: *ν* stretching, *δ* bending, *s* symmetric, *as* asymmetric vibrations and *out* out-plane; (Δ) are compared to controls in frequency shift, “↑” in the column of (Δ) represents blue-shift: moving towards high wave number, “↓” in the column of (Δ) represents red-shift: moving to a lower wave-number direction; Changes (%) are compared to controls in average band absorbance, “↑” in the column of (%) represents increased change and “↓” in the column of (%) represents decreased change. The details of the spectral change are depicted in Appendix A.

**Table 3 biomolecules-12-01815-t003:** Comparison of classification accuracies of methods based on different feature datasets.

Methods	Accuracies of Classification		
	Raw IR Data (%)	SD-IR Data (%)	Combined Data(%)
BP	72.3 (±0.31)	91.4 (±0.24)	97.1 (±0.11)
KNN	81.1 (±0.14)	85.6 (±0.09)	93.8 (±0.06)
RF	76.3 (±0.18)	87.0 (±0.13)	92.7 (±0.03)
DT	73.0 (±0.26)	84.5 (±0.14)	92.7 (±0.19)
Logistic	71.1 (±0.29)	85.6 (±0.26)	96.6 (±0.06)
SVM	74.5 (±0.15)	82.4 (±0.13)	97.7 (±0.19)
MVLR	87.5 (±0.09)	96.4 (±0.04)	100.0 (±0)
PLS-DA	88.2 (±0.05)	97.9 (±0.02)	100.0 (±0)

Backpropagation (BP) neural network, K-nearest Neighbor (KNN), random forest (RF), decision tree (DT), logistic regression, support vector machine (SVM), multiple linear regression (MVLR), and partial least squares discriminant analysis (PLS-DA). % are classification accuracies (standard deviation).

## Data Availability

All data are contained within the manuscript and available upon request.

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
