# Peer review of "Fast and Deep Diagnosis Using Blood-Based ATR-FTIR Spectroscopy for Digestive Tract Cancers"

_biomolecules, 2022, doi:10.3390/biom12121815_

Round 1
Reviewer 1 Report
The work has been written in a very detailed way and the proposed analysis methodologies are well described. The results were presented in depth. The discussion seems to me a bit narrow in comparison to the amount of results presented. Tables 2 and 3 should be placed in a clearer and more readable way. Overall I think it is a very good job.
Author Response
Response to 'Comment on " Fast and deep diagnosis using blood-based ATR-FTIR spectroscopy for digestive tract cancers "
We would like to thank the reviewer for the constructive comments. These comments are well taken by us, and we have made responses and changes accordingly. The following is the point-by-point responses to the reviewer's comments, and the changes, correspondingly based on the point-by-point responses to reviewer, were clearly marked by the blue color in our revised manuscript.
Reviewer's Comments to Author:
Reviewer #1:
The work has been written in a very detailed way and the proposed analysis methodologies are well described. The results were presented in depth. The discussion seems to me a bit narrow in comparison to the amount of results presented. Tables 2 and 3 should be placed in a clearer and more readable way. Overall I think it is a very good job.
Response: Thank you so much for your suggestions. We have taken to your suggestions and adjusted Tables 2 and 3 in our revised manuscript in a clearer and more readable way.
Table 2. Comparison and connection of clinic biochemistry and FTIR measurement for serum samples of DTCs.
Serum biochemistry |
Structural formula |
Function groups |
LC |
GC |
CC |
||||||
Test Item |
Reference range |
Results |
FTIR Band (Δ) |
Absorb (%) |
FTIR Band (Δ) |
Absorb (%) |
FTIR Band (Δ) |
Absorb (%) |
|||
CEA |
0-5 ng/ml |
LC: 1.37 (±0.07) GC: 34.83 (±0.12) CC: 68.70 (±0.25) |
|
δ(NH)
ν(C-O-H) |
1508.1↑
1041.9↑ |
0.74↑
0.68↑ |
1508.0↑
1039.7 |
0.72↑
0.69↑ |
1508.1↑
1041.2↑ |
0.73↑
0.68 |
|
CA |
0-35 U/ml |
LC: 54.74 (±0.21) GC: 201.02 (±0.63) CC: 423.79 (±0.47) |
νs(NH-C=O)
δout(NH) |
1119.4↑
984.8↑ |
0.68↑
0.70↑ |
1119.4↑
986.1↑ |
0.67↑
0.69↑ |
1118.2↑
986.1↑ |
0.66↑
0.70↑ |
||
AFP |
0-9 ng/ml |
LC: 701.01 (±0.52) |
ν(C=O) |
1638.2↓ |
0.02↑ |
|
|
|
|
||
Liver cancer (LC), gastric cancer (GC), colorectal cancer (CC). Carcinoembryonic antigen (CEA), carcinoembryonic antigen (CA), alpha-fetoprotein (AFP). Results are in mean (standard deviation); major molecular vibrations: ν stretching, δ bending, s symmetric, as asymmetric vibrations and out out-plane; (Δ) are compared to controls in frequency shift, "↑" in the column of (Δ) represents blue-shift: moving towards high wave number, "↓" in the column of (Δ) represents red-shift: moving to a lower wave-number direction; Changes (%) are compared to controls in average band absorbance, "↑" in the column of (%) represents increased change and "↓" in the column of (%) represents decreased change. The details of spectral change are depicted in Supplementary Table S2. |
|||||||||||
Table 3. Comparison of classification accuracies of methods based on different feature datasets.
Methods |
Accuracies of classification |
||
Raw IR data (%) |
SD-IR data (%) |
Combined data(%) |
|
BP KNN RF DT Logistic SVM |
72.3 (±0.31) 81.1 (±0.14) 76.3 (±0.18) 73.0 (±0.26) 71.1 (±0.29) 74.5 (±0.15) |
91.4 (±0.24) 85.6 (±0.09) 87.0 (±0.13) 84.5 (±0.14) 85.6 (±0.26) 82.4 (±0.13) |
97.1 (±0.11) 93.8 (±0.06) 92.7 (±0.03) 92.7 (±0.19) 96.6 (±0.06) 97.7 (±0.19) |
MVLR PLS-DA |
87.5 (±0.09) 88.2 (±0.05) |
96.4 (±0.04) 97.9 (±0.02) |
100.0 (±0) 100.0 (±0) |
Back propagation (BP) neural network, K-nearest Neighbor (KNN), random forest (RF), decision tree (DT), and logistic regression, support vector machine (SVM), multiple linear regression (MVLR), and partial least squares discriminant analysis (PLS-DA). % are classification accuracies (standard deviation). |

Reviewer 2 Report
It is good idea of 2D-SD-IR dataset for the diagnosis of cancers. But the FT-IR spectra were very small changes, which will be difficult to take the diagnosis of these cancers and the spectral components will not so clear. Accordingly, it will be difficult to accept it in this journal.
Author Response
Response to 'Comment on " Fast and deep diagnosis using blood-based ATR-FTIR spectroscopy for digestive tract cancers "
We would like to thank the reviewer for the constructive comments. These comments are well taken by us, and we have made responses and changes accordingly. The following is the point-by-point responses to the reviewer's comments, and the changes, correspondingly based on the point-by-point responses to reviewer, were clearly marked by the blue color in our revised manuscript.
Reviewer #2:
It is good idea of 2D-SD-IR dataset for the diagnosis of cancers. But the FT-IR spectra were very small changes, which will be difficult to take the diagnosis of these cancers and the spectral components will not so clear. Accordingly, it will be difficult to accept it in this journal.
Response: Thank you. Actually, compared to traditional transmission-FTIR for dehydrated samples with significant absorbance IR spectral changes and an influential spectral shape, the in-situ IR spectra of fresh blood samples, without dehydration and 3μL per sample, were measured by attenuated total reflection mode of FTIR (ATR-FTIR) with high brilliance synchrotron light source for reducing water-induced absorption and improving single-to-noise rate (SNR) to obtain in-situ and reliable spectra of DTCs. It seemed that there were small changes of original absorbance spectra via average and smooth processing, and the changes were not uniform during the range of 900-3500 cm-1, seen in Figure 2. However, benefitted from adequate SNR, there were more abundant and significant quantitative variations of fine band locations and relative absorbance in in-situ SD-IR spectra between the dataset of IMF band in the range of 1200-1700 cm-1 and control groups (p<0.001, Supplementary Table S2), suggesting definite transitions in protein secondary structures and significant changes of functional groups. Among them, the p value was used to evaluate the statistical spectral differences of major bands in absorbance of DTC patients and healthy controls and p was considered significant at 0.05.
Supplementary Table S2. SD-IR spectra: Distribution and statistical comparison of DTC and control spectra in terms of band position and relative absorbance.
Centered bands |
Band locations/(cm-1) |
|
Absorbance (a.u.) |
||||
LC (p) |
GC (p) CC (p) |
|
LC (%) |
GC (%) |
CC (%) |
||
3195 3089 2965 2925 2898 2857 1741 1695 1639 1574 1544 1508 1455 1427 1398 1354 1309 1240 1166 1117 1082 1040 986 |
3195.8 3089.3(*) 2967.9(***) 2925.9 2898.4(*) 2855.7(**) 1742.6(**) 1693.4(**) 1638.2(**) 1574.9(**) 1545.9(*) 1508.1(**) 1456.0(*) 1428.2(*) 1399.6(*) 1354.2 1310.7(**) 1242.1(***) 1165.2(*) 1119.4(***) 1082.6(***) 1041.9(***) 984.8 |
3196.8(**) 3089.1(*) 2965.5(*) 2924.7(**) 2899.0 2856.2(*) 1740.6 1698.9(***) 1639.7 1574.1(**) 1544.1(*) 1508(**) 1456.3(*) 1426.5(*) 1398.6 1354.1 1309.7(*) 1239.4 1165.2(*) 1119.4(***) 1080.8(**) 1039.7 986.1(**) |
3195.1(*) 3088.8 2966.7(**) 2925.9 2897.7(**) 2859.2(***) 1741.6(**) 1695.3(***) 1638.6(**) 1575.6(*) 1544.9(*) 1508.1(**) 1455.3 1427.3 1399.6(*) 1354.3 1310.7 1241.1(**) 1166.2 1118.2(***) 1082.7(***) 1041.2(**) 986.1(**) |
0.53(0) 0.66(0) 0.68(+4.6) 0.64(+8.5) 0.72(+1.4) 0.66(+3.1) 0.69(+3.0) 0.84(-2.3) 0.02(+100.0) 0.85(-5.6) 0.53(+26.2) 0.74(+4.2) 0.63(+3.3) 0.72(-2.7) 0.64(+6.7) 0.69(+1.5) 0.68(+6.3) 0.67(+4.7) 0.67(+1.5) 0.67(0) 0.69(+3.0) 0.68(+3.0) 0.70(+2.9) |
0.56(+5.6) 0.66(0) 0.65(0) 0.62(+5.0) 0.73(+2.8) 0.63(-1.6) 0.68(+1.5) 0.84(-2.3) 0.07(+600.0) 0.88(-2.2) 0.41(-2.4) 0.72(+1.4) 0.60(-1.6) 0.75(+1.3) 0.57(-5.0) 0.68(0) 0.65(+1.6) 0.66(+3.1) 0.66(0) 0.67(0) 0.65(-3.0) 0.69(+4.5) 0.69(+1.5) |
0.55(+3.7) 0.67(+1.5) 0.68(+4.6) 0.66(+11.9) 0.71(0) 0.67(+4.7) 0.71(+6.0) 0.84(-2.3) 0.02(+100.0) 0.88(-2.2) 0.53(+26.2) 0.73(+2.8) 0.63(+3.3) 0.73(-1.3) 0.64(+6.7) 0.69(+1.5) 0.67(+4.7) 0.67(+4.7) 0.67(1.5) 0.67(0) 0.68(+1.5) 0.68(+3.0) 0.70(+2.9) |
|
Tumors: Liver cancer (LC), gastric cancer (GC) and colorectal cancer (CC); Changes (%) are compared to controls in average band absorbance; *p<0.05, **p<0.01, ***p<0.001. |
|||||||
Based on the quantitative analysis of in-situ SD-IR dataset, including significant changes of absorbance and peak positions of biomacromolecules, such as lipid, protein, saccharide, and nucleic acid compositions, it’s possible to build a multi-feature auto-classification protocol for subtypes and stages of DTCs. Therefore, the construction of in-situ 2D-SD-IR-based IR feature dataset for comprehensive diagnosis of DTCs can be combined with different data mining (DM) methods to explore a most suitable diagnosis pattern of DCT. Using the selected DM methods such as PLS-DA and BP ML classifiers, the identifications of inter-group and intra-group (stages) of DTCs achieved the following performance of sensitivity and specificity (Table 3, Figure 4).
Table 3. Comparison of classification accuracies of methods based on different feature datasets.
Methods |
Accuracies of classification |
||
Raw IR data (%) |
SD-IR data (%) |
Combined data(%) |
|
BP KNN RF DT Logistic SVM |
72.3 (±0.31) 81.1 (±0.14) 76.3 (±0.18) 73.0 (±0.26) 71.1 (±0.29) 74.5 (±0.15) |
91.4 (±0.24) 85.6 (±0.09) 87.0 (±0.13) 84.5 (±0.14) 85.6 (±0.26) 82.4 (±0.13) |
97.1 (±0.11) 93.8 (±0.06) 92.7 (±0.03) 92.7 (±0.19) 96.6 (±0.06) 97.7 (±0.19) |
MVLR PLS-DA |
87.5 (±0.09) 88.2 (±0.05) |
96.4 (±0.04) 97.9 (±0.02) |
100.0 (±0) 100.0 (±0) |
Back propagation (BP) neural network, K-nearest Neighbor (KNN), random forest (RF), decision tree (DT), and logistic regression, support vector machine (SVM), multiple linear regression (MVLR), and partial least squares discriminant analysis (PLS-DA). % are classification accuracies (standard deviation). |
Figure 4. Confusion matrix of the prediction accuracy for SD-IR-based blood of DTC stages. (a-c) Prediction reports for the SD-IR absorbance feature dataset from patients with DTCs. (d-f) Prediction results of the optimized SD-IR absorbance + shift feature dataset. The 2D-SD-IR-BP algorithm exhibited best performance. 1-4: represent detailed staging of patients with DTCs from Supplementary Table S1, respectively.
Thus, the proposed in-situ 2D-SD-IR-BP diagnostic protocol, honored to have been appreciated by you, exhibited excellent performance for different aspects of diagnosis related to DTCs and demonstrated that (a) in-situ blood-based ATR-FTIR spectra is essential to build 2D feature dataset for machine learning, avoiding uselessness of wavenumber shifting distorted by dehydration, (b) IMFs in the range of 1700-1400 cm-1 are effectively measurable by ATR-FTIR and provide abundant specific diagnostic information, (c) according to similarity of DTCs, the PLS-DA and BP ML methods were optimized respectively for inter-group and intra-group classification, (d) SD-IR data is superior to IR data for extraction and utilization of small bio-macromolecular changes. Therefore, the findings are helpful for exploration of auto-identification feasibility for different level diagnosis of DTC.

Reviewer 3 Report
My remarks:
1. The authors cite a sufficient number of interesting and useful papers, but often the references do not match the citations. For example, ref. 2, 3, 4, 6, 32.
1)Line 80. The authors write about Synchrotron radiation (SR) light source and refer to papers [27, 28, 29]. These references are useful for the manuscript topic, but they do not contain information about the source of synchrotron radiation.
2)Line 184-185. The authors write: ”HCA and PCA are widely used unsupervised approaches in microorganism differentiation [33-35]. ([33]Fogarty, S.W.; Patel, II; Trevisan, J.; Nakamura, T.; Hirschmugl, C.J.; Fullwood, N.J.; Martin, F.L. Sub-cellular spectrochemical imaging of isolated human corneal cells employing synchrotron radiation-based Fourier-transform infrared microspectroscopy. Analyst 2013, 138, 240-248, doi:10.1039/c2an36197c. [34]. Ballabio, D. & Consonni, V. Classification tools in chemistry. Part 1: linear models. PLS-DA. Analytical Methods 2013, 5, doi:10.1007/s00216-018-1111-x. [35]. Patel, II; Shearer, D.A.; Fogarty, S.W.; Fullwood, N.J.; Quaroni, L.; Martin, F.L.; Weisz, J. Infrared microspectroscopy identifies biomolecular changes associated with chronic oxidative stress in mammary epithelium and stroma of breast tissues from healthy young women: implications for latent stages of breast carcinogenesis. Cancer Biol Ther 2014, 15, 225-235, 523 doi:10.4161/cbt.26748). - As you can see, these papers do not contain data on microorganism differentiation.
3) [29] Zhang, L.; Xiao, M.; Wang, Y.; Peng, S.; Chen, Y.; Zhang, D.; Zhang, D.; Guo, Y.; Wang, X.; Luo, H.; et al. Fast Screening and 505 Primary Diagnosis of COVID-19 by ATR-FT-IR. Anal Chem 2021, 93, 2191-2199, doi:10.1021/acs.analchem.0c04049. This reference corresponds to the reference [30] in the text.
Probably, there was a shift in references during editing. This needs to be carefully checked.
2. Line 60-64. This sentence is too long and difficult to understand. The second parenthesis is missing.
3. Table 1 does not show data for control samples.
4. Line 320. The authors write: “then the most discriminating bands located at 1683, 1640, 1514, 1086, and 1026 cm-1, listed in the SD-IR spectra in Supplementary Table S2. “ But in the table S2 we can see the frequencies:1695, 1639, 1508, 1082, and 1040 cm-1 . Why?
Author Response
Response to 'Comment on " Fast and deep diagnosis using blood-based ATR-FTIR spectroscopy for digestive tract cancers "
We would like to thank the reviewer for the constructive comments. These comments are well taken by us, and we have made responses and changes accordingly. The following is the point-by-point responses to the reviewer's comments, and the changes, correspondingly based on the point-by-point responses to reviewer, were clearly marked by the blue color in our revised manuscript.
Reviewer's Comments to Author:
Reviewer #3:
- The authors cite a sufficient number of interesting and useful papers, but often the references do not match the citations. For example, ref. 2, 3, 4, 6, 32.
Response: Thank you so much for your suggestions. We apologized for our impertinences and revised the references in our revised manuscript as following: “With the extensive or excessive use of endoscopy, CT & MRI, the diagnosis rate of medium and advanced DTCs has been significantly increased [2,3], however, the 5-year survival rate of DTC patients has not distinctly improved over the decades [4,5].” and ”Analysis of variance (ANOVA) [31] is a well-established method to evaluate the difference of the observed statistics among groups by calculating the F-statistic.”
References:
- Yan, M.; Wang, W. Radiomic Analysis of CT Predicts Tumor Response in Human Lung Cancer with Radiotherapy. J Digit Imaging 2020, 33, 1401-1403, doi:10.1007/s10278-020-00385-3.
- Ren, Y.; Cao, Y.; Hu, W.; Wei, X.; Shen, X. Diagnostic accuracy of computed tomography imaging for the detection of differences between peripheral small cell lung cancer and peripheral non-small cell lung cancer. Int J Clin Oncol 2017, 22, 865-871, doi:10.1007/s10147-017-1131-0.
- Siegel, R.L.; Miller, K.D.; Fuchs, H.E.; Jemal, A. Cancer statistics, 2022. CA Cancer J Clin 2022, 72, 7-33, doi:10.3322/caac.21708.
- Cao, W.; Chen, H.D.; Yu, Y.W.; Li, N.; Chen, W.Q. Changing profiles of cancer burden worldwide and in China: a secondary analysis of the global cancer statistics 2020. Chin Med J (Engl) 2021, 134, 783-791, doi:10.1097/CM9.0000000000001474.
- Guleken, Z.; Bulut, H.; Gultekin, G.I.; Arikan, S.; Yaylim, I.; Hakan, M.T.; Sonmez, D.; Tarhan, N.; Depciuch, J. Assessment of structural protein expression by FTIR and biochemical assays as biomarkers of metabolites response in gastric and colon cancer. Talanta 2021, 231, 122353, doi:10.1016/j.talanta.2021.122353.
1)Line 80. The authors write about Synchrotron radiation (SR) light source and refer to papers [27, 28, 29]. These references are useful for the manuscript topic, but they do not contain information about the source of synchrotron radiation.
Response: Thank you so much for your suggestions. We apologized for our impertinences and revised the references in our revised manuscript as following: “Synchrotron radiation (SR) light source has the advantage of brightness at least two or three orders of magnitude higher than traditional light source (globar), with improved spectral signal-to-noise ratio (SNR) of samples [26]. Thus, the SR-based ATR-FTIR microscopy facilitates the high-precision in-situ measurement of fresh samples, especially achieving specific spectrum collections of microregion-of-interest [27].”
References:
- Xie, H.; Deng, B.; Du, G.; Fu, Y.; He, Y.; Guo, H.; Peng, G.; Xue, Y.; Zhou, G.; Ren, Y.; et al. X-ray biomedical imaging beamline at SSRF. Journal of Instrumentation 2013, 8, C08003-C08003, doi:10.1088/1748-0221/8/08/c08003.
- Guo, S.; Xiu, J.; Kong, L.; Kong, X.; Wang, H.; Lü, Z.; Xu, F.; Li, J.; Ji, T.; Wang, F.; Liu, H. Micro-tomographic and infrared spectral data mining for breast cancer diagnosis. Opt Laser Eng 2022, 160, 107305, doi: 10.1016/j.optlaseng.2022.107305.
2)Line 184-185. The authors write: ”HCA and PCA are widely used unsupervised approaches in microorganism differentiation [33-35]. ([33]Fogarty, S.W.; Patel, II; Trevisan, J.; Nakamura, T.; Hirschmugl, C.J.; Fullwood, N.J.; Martin, F.L. Sub-cellular spectrochemical imaging of isolated human corneal cells employing synchrotron radiation-based Fourier-transform infrared microspectroscopy. Analyst 2013, 138, 240-248, doi:10.1039/c2an36197c. [34]. Ballabio, D. & Consonni, V. Classification tools in chemistry. Part 1: linear models. PLS-DA. Analytical Methods 2013, 5, doi:10.1007/s00216-018-1111-x. [35]. Patel, II; Shearer, D.A.; Fogarty, S.W.; Fullwood, N.J.; Quaroni, L.; Martin, F.L.; Weisz, J. Infrared microspectroscopy identifies biomolecular changes associated with chronic oxidative stress in mammary epithelium and stroma of breast tissues from healthy young women: implications for latent stages of breast carcinogenesis. Cancer Biol Ther 2014, 15, 225-235, 523 doi:10.4161/cbt.26748). - As you can see, these papers do not contain data on microorganism differentiation.
Response: Thank you so much for your suggestions. We apologized for the lack of clarity of expression and revised in our revised manuscript as following: “HCA and PCA are widely used unsupervised approaches for analyzing data.”
3) [29] Zhang, L.; Xiao, M.; Wang, Y.; Peng, S.; Chen, Y.; Zhang, D.; Zhang, D.; Guo, Y.; Wang, X.; Luo, H.; et al. Fast Screening and 505 Primary Diagnosis of COVID-19 by ATR-FT-IR. Anal Chem 2021, 93, 2191-2199, doi:10.1021/acs.analchem.0c04049. This reference corresponds to the reference [30] in the text.
Response: Thank you so much for your suggestions. We apologized for our misquotation and revised the references in our revised manuscript as following:” With the combination of data mining (DM) methods, such as principal component analysis (PCA), Hierarchical cluster analysis (HCA), Partial least squares discriminant analysis (PLS-DA), and machine learning (ML) classifiers, the FTIR-based intelligent diagnosis has been widely employed into Corona Virus Disease 2019 (COVID-19) screening [28], health monitoring [29], disease detection [30], and food security.”
References:
- Zhang, L.; Xiao, M.; Wang, Y.; Peng, S.; Chen, Y.; Zhang, D.; Zhang, D.; Guo, Y.; Wang, X.; Luo, H.; et al. Fast Screening and Primary Diagnosis of COVID-19 by ATR-FT-IR. Anal Chem 2021, 93, 2191-2199, doi:10.1021/acs.analchem.0c04049.
- Huber, M.; Kepesidis, K.V.; Voronina, L.; Bozic, M.; Trubetskov, M.; Harbeck, N.; Krausz, F.; Zigman, M. Stability of per-son-specific blood-based infrared molecular fingerprints opens up prospects for health monitoring. Nat Commun 2021, 12, 1511, doi:10.1038/s41467-021-21668-5.
- Bonnier, F.; Blasco, H.; Wasselet, C.; Brachet, G.; Respaud, R.; Carvalho, L.F.; Bertrand, D.; Baker, M.J.; Byrne, H.J.; Chourpa, I. Ultra-filtration of human serum for improved quantitative analysis of low molecular weight biomarkers using ATR-IR spectroscopy. Analyst 2017, 142, 1285-1298, doi:10.1039/c6an01888b.
- Line 60-64. This sentence is too long and difficult to understand. The second parenthesis is missing.
Response: Thank you so much for your suggestions. We apologized for our ambiguous expressions and revised in our revised manuscript as following: “The Fourier-transform infrared spectroscopy (FTIR) is widely used in cancer screening, characterized as a simple, label free, non-invasive technique. Many potential IMFs for disease diagnostics have been reported (e.g., brain [14], lung [15], prostate [16], ovarian [17] and breast [18] cancers; metabolic changes related to health monitoring [19,20]).”
References:
- Butler, H.J.; Brennan, P.M.; Cameron, J.M.; Finlayson, D.; Hegarty, M.G.; Jenkinson, M.D.; Palmer, D.S.; Smith, B.R.; Baker, M.J. Development of high-throughput ATR-FTIR technology for rapid triage of brain cancer. Nat Commun 2019, 10, 4501, doi:10.1038/s41467-019-12527-5.
- Kong, X.; Wang, F.; Guo, S.; Wang, H.; Lü, Z.; Xu, C.; Guan, Y.; Kong, L.; Li, J.; Wei, G.; et al. Structural and spectral morphometry and diagnosis of lung tumors. Infrared Physics & Technology 2022, 124, doi:10.1016/j.infrared.2022.104229.
- Ollesch, J.; Theegarten, D.; Altmayer, M.; Darwiche, K.; Hager, T.; Stamatis, G.; Gerwert, K.; Gerwert, K. An infrared spectroscopic blood test for non-small cell lung carcinoma and subtyping into pulmonary squamous cell carcinoma or adenocarcinoma. Biomedical Spectroscopy and Imaging 2016, 5, 129-144, doi:10.3233/bsi-160144.
- Gajjar, K.; Trevisan, J.; Owens, G.; Keating, P.J.; Wood, N.J.; Stringfellow, H.F.; Martin-Hirsch, P.L.; Martin, F.L. Fourier-transform infrared spectroscopy coupled with a classification machine for the analysis of blood plasma or serum: a novel diagnostic approach for ovarian cancer. Analyst 2013, 138, 3917-3926, doi:10.1039/c3an36654e.
- Zelig, U.; Barlev, E.; Bar, O.; Gross, I.; Flomen, F.; Mordechai, S.; Kapelushnik, J.; Nathan, I.; Kashtan, H.; Wasserberg, N.; et al. Early detection of breast cancer using total biochemical analysis of peripheral blood components: a preliminary study. BMC Cancer 2015, 15, 408, doi:10.1186/s12885-015-1414-7.
- Krafft, C.; Wilhelm, K.; Eremin, A.; Nestel, S.; von Bubnoff, N.; Schultze-Seemann, W.; Popp, J.; Nazarenko, I. A specific spectral signature of serum and plasma-derived extracellular vesicles for cancer screening. Nanomedicine 2017, 13, 835-841, doi:10.1016/j.nano.2016.11.016.
- Sahu, R.K.; Zelig, U.; Huleihel, M.; Brosh, N.; Talyshinsky, M.; Ben-Harosh, M.; Mordechai, S.; Kapelushnik, J. Continuous monitoring of WBC (biochemistry) in an adult leukemia patient using advanced FTIR-spectroscopy. Leuk Res 2006, 30, 687-693, doi:10.1016/j.leukres.2005.10.011.
- Table 1 does not show data for control samples.
Response: Thank you so much for your suggestions. We apologized for the missing data for the control sample and adjusted Table 1 in our revised manuscript.
Table 1. Standardized spectra: Distribution and statistical comparison of DTC and control spectra in terms of band position and relative absorbance.
Assignments |
Band locations/(cm-1) |
|
Relative absorbance (a.u.) |
|
|||||||
LC (p) |
GC(p) CC (p) |
Control |
LC |
GC |
CC |
Control |
|||||
amide A νas (CH3) νas (CH2) amide I amide II δ(CH2) νs (COO-) amide III νas (PO2-) νs (C-O-C) νs (PO2-) |
3366.2 (**) 2962.4 (***) 2927.8 (**) 1637.5 1548.8 (**) 1456.2 (***) 1400.0 (***) 1315.4 (***) 1245.9 (**) 1168.8 (**) 1089.7 (**) |
3364.2 2956.7 (**) 2925.8 1639.4 (**) 1544.9 (**) 1455.6 (***) 1398.3 (**) 1309.6 (**) 1247.9 1164.9 (***) 1083.9 (***) |
3364.2 2960.6 (***) 2923.9 (**) 1638.2 (*) 1554.5 (***) 1460.0 (***) 1402.2 (***) 1313.4 (***) 1244.0 (***) 1167.3 (*) 1081.8 (***) |
3346.3 2954.8 2925.8 1637.6 1546.9 1452.3 1397.6 1311.5 1247.2 1166.9 1091.6 |
1.00 1.06 1.00 1.00 0.98 1.11 0.95 0.93 0.87 0.83 0.92 |
1.00 1.12 1.06 1.00 1.04 1.05 1.05 1.26 1.00 1.00 1.00 |
1.00 0.94 0.82 1.00 0.79 0.83 0.73 0.73 0.67 0.67 0.58 |
1.00 1.07 0.83 1.00 1.03 0.83 1.02 1.02 0.91 0.91 0.91 |
|||
|
|
Liver cancer (LC), gastric cancer (GC), colorectal cancer (CC). ν stretching, δ bending, s symmetric, as asymmetric vibrations. Statistical differences were compared between DTCs and the control group using ANOVA. *p<0.05, **p<0.01, ***p<0.001. |
|
||||||||
- Line 320. The authors write: “then the most discriminating bands located at 1683, 1640, 1514, 1086, and 1026 cm-1, listed in the SD-IR spectra in Supplementary Table S2. “ But in the table S2 we can see the frequencies:1695, 1639, 1508, 1082, and 1040 cm-1 . Why?
Response: Thank you so much for your suggestions. We apologized for our ambiguous expressions. The wavenumbers of 1695, 1639, 1508, 1082 and 1040 cm-1 in Table S2 represent the mean central bands of LC, GC and CC. We have revised in our revised manuscript as following: “Subsequently, it indicated that VIP > 1 preferred to assess the impact of each single variable in the model and then the most discriminating bands of LC located at 1683, 1640, 1514, 1086, and 1026 cm-1. Similarly, the most discriminative frequency bands of GC located at 1686, 1655, 1563, 1515, 1356 and 1202 cm-1. The band of CC located at 1773, 1742, 1714, 1470, 1062 and 1038 cm-1. This finding is probably consistent with the sub-peaks correspondence in the SD-IR spectrum of LC, GC and CC as shown in Supplementary Table S2.”

Reviewer 4 Report
The authors suggested and investigated a method for digestive tract cancers diagnostics using blood-based ATR-FTIR spectroscopy and machine learning. The material is quite interesting for specialists in biophotonics and molecular medicine.
Several comments.
1. The description of the groups under study (lines 148-152) is repeated in lines 210-212.
2. The authors used 10-fold cross-validation (line 300) but data about accuracy of analyzed prediction data models are presented in Table 3 (and in Table s3) by one (mean -?) value. These estimations should be presented in term of mean and StD values. Seems to be, here, the authors used a full DTC data set without taking into account that this dataset consists of three DTC subgroups. How do they divide three subgroups between training and testing datasets?
3. Figure 3 should be described in more detail.
4. In the section 3.4., there is no information about the method of cross-validation and DTC subgroups split.
Author Response
Response to 'Comment on " Fast and deep diagnosis using blood-based ATR-FTIR spectroscopy for digestive tract cancers "
We would like to thank the reviewer for the constructive comments. These comments are well taken by us, and we have made responses and changes accordingly. The following is the point-by-point responses to the reviewer's comments, and the changes, correspondingly based on the point-by-point responses to reviewer, were clearly marked by the blue color in our revised manuscript.
Reviewer's Comments to Author:
Reviewer #4:
The authors suggested and investigated a method for digestive tract cancers diagnostics using blood-based ATR-FTIR spectroscopy and machine learning. The material is quite interesting for specialists in biophotonics and molecular medicine.
Several comments.
- The description of the groups under study (lines 148-152) is repeated in lines 210-212.
Response: Thank you so much for your suggestions. We apologized for the repetition of our statements and removed the repetitive phrases in our revised manuscript.
- The authors used 10-fold cross-validation (line 300) but data about accuracy of analyzed prediction data models are presented in Table 3 (and in Table s3) by one (mean -?) value. These estimations should be presented in term of mean and StD values. Seems to be, here, the authors used a full DTC data set without taking into account that this dataset consists of three DTC subgroups. How do they divide three subgroups between training and testing datasets?
Response: Thank you so much for your suggestions. We apologized for the lack of clarity in data presentation and have added the mean and SD values for classification accuracy in Table 3 and Table S3.
In this study, ML techniques are applied to a spectral dataset. The purpose of this approach is to identify the signals of cancer from a known patient cohort to develop a trained classification model, and then to use this information to predict the presence of cancer in an unknown population. The supervised classification methods and different datasets of ATR-FTIR were employed and compared to choose the best pathway of diagnostic feasibility. Each dataset was subsequently downscaled by PCA, 70% training and 30% test sets were created during each iteration, and the prediction performances were estimated by applying 10-fold cross-validation. The process uses labeled data, so that the three subgroups divided between the training and test datasets are known.
Table 3. Comparison of classification accuracies of methods based on different feature datasets.
Methods |
Accuracies of classification |
||
Raw IR data (%) |
SD-IR data (%) |
Combined data(%) |
|
BP KNN RF DT Logistic SVM |
72.3 (±0.31) 81.1 (±0.14) 76.3 (±0.18) 73.0 (±0.26) 71.1 (±0.29) 74.5 (±0.15) |
91.4 (±0.24) 85.6 (±0.09) 87.0 (±0.13) 84.5 (±0.14) 85.6 (±0.26) 82.4 (±0.13) |
97.1 (±0.11) 93.8 (±0.06) 92.7 (±0.03) 92.7 (±0.19) 96.6 (±0.06) 97.7 (±0.19) |
MVLR PLS-DA |
87.5 (±0.09) 88.2 (±0.05) |
96.4 (±0.04) 97.9 (±0.02) |
100.0 (±0) 100.0 (±0) |
Back propagation (BP) neural network, K-nearest Neighbor (KNN), random forest (RF), decision tree (DT), and logistic regression, support vector machine (SVM), multiple linear regression (MVLR), and partial least squares discriminant analysis (PLS-DA). % are classification accuracies (standard deviation). |
Supplementary Table S3. Accuracy of results obtained using multiple multivariate methods.
Cancers |
Methods |
Accuracies of classification |
|||
IR Absorbance(%) |
IR Absorbance +shift(%) |
SD-IR Absorbance(%) |
SD-IR Absorbance +shift(%) |
||
LC |
BP KNN RF DT Logistic SVM MVLR PLS-DA |
70.60 (±0.44) 81.75 (±0.33) 75.00 (±0.41) 83.75 (±0.25) 65.00 (±0.49) 68.75 (±0.36) 78.86 (±0.24) 82.30 (±0.16) |
89.40 (±0.23) 89.85 (±0.18) 83.75 (±0.22) 89.87 (±0.14) 75.00 (±0.22) 86.25 (±0.21) 82.43 (±0.19) 87.20 (±0.12) |
76.90 (±0.23) 83.87 (±0.12) 80.65 (±0.17) 88.75 (±0.12) 70.00 (±0.21) 86.87 (±0.35) 87.40 (±0.23) 88.25 (±0.21) |
95.60 (±0.06) 93.00 (±0.09) 86.25 (±0.13) 92.50 (±0.07) 83.25 (±0.12) 94.75 (±0.02) 89.20 (±0.13) 91.47 (±0.05) |
GC |
BP KNN RF DT Logistic SVM MVLR PLS-DA |
75.00 (±0.22) 78.75 (±0.23) 72.25 (±0.31) 74.30 (±0.26) 58.75 (±0.45) 46.88 (±0.49) 77.20 (±0.11) 79.50 (±0.13) |
85.20 (±0.15) 79.00 (±0.28) 78.75 (±0.31) 84.13 (±0.24) 85.00 (±0.12) 76.25 (±0.29) 84.25 (±0.17) 85.20 (±0.15) |
77.50 (±0.33) 82.50 (±0.25) 75.63 (±0.21) 83.00 (±0.15) 80.62 (±0.14) 84.50 (±0.12) 86.45 (±0.09) 92.20 (±0.01) |
98.70 (±0.06) 96.87 (±0.02) 88.13 (±0.12) 92.50 (±0.09) 89.25 (±0.11) 93.12 (±0.07) 94.75 (±0.02) 98.42 (±0.01) |
CC |
BP KNN RF DT Logistic SVM MVLR PLS-DA |
65.60 (±0.33) 69.37 (±0.29) 61.80 (±0.38) 65.00 (±0.33) 56.87 (±0.41) 55.00 (±0.43) 69.50 (±0.28) 70.50 (±0.14) |
70.60 (±0.24) 72.50 (±0.27) 69.87 (±0.29) 73.12 (±0.26) 68.12 (±0.31) 56.25 (±0.34) 74.50 (±0.20) 76.30 (±0.17) |
69.40 (±0.24) 74.87 (±0.28) 81.25 (±0.25) 78.87 (±0.27) 68.13 (±0.32) 79.65 (±0.23) 80.74 (±0.19) 82.20 (±0.18) |
95.00 (±0.12) 90.25 (±0.02) 91.00 (±0.12) 92.50 (±0.10) 78.75 (±0.21) 96.85 (±0.05) 91.13 (±0.09) 92.75 (±0.04) |
- Figure 3 should be described in more detail.
Response: Thank you so much for your suggestions. We apologized for the lack of clarity of expression and added the statement about Figure 3 in our revised manuscript as following:” Hence, we adopted the PLS-DA method to identification of different DTC types with all serum samples of DTCs (Figure 3). The latent variables (LV) of PLS-DA with cross-validation were down-dimensioned by LV scores into three principled variables, and then R2Y and Q2 values were used to evaluate the fitting effect of the PLS-DA model. R2Y denotes the percentage of Y-matrix information that can be explained by the PLS-DA classification model, and Q2 is to evaluate the predictive ability of the PLS-DA model. They were evaluated by R2Y values (0.991, 0.977, and 0.997) and Q2 values (0.775, 0.752, and 0.781) respectively. Generally, R2Y close to 1 and Q2 > 0.5 is recognized as well-predictive of model predictability. The regression vectors of each selected variable (LV loadings) and the importance of the corresponding variables were visualized in pseudo-color based on the VIP in Figure 3 (a). The VIP scores were shown in Supplementary Figure S3 (a) and evaluated by the prediction error rates using three cross-validation methods, all exhibiting good predictability of this model (Supplementary Figure S3 (b)). Subsequently, it indicated that VIP > 1 preferred to assess the impact of each single variable in the model and then the most discriminating bands of LC located at 1683, 1640, 1514, 1086, and 1026 cm-1. Similarly, the most discriminative frequency bands of GC located at 1686, 1655, 1563, 1515, 1356 and 1202 cm-1. The band of CC located at 1773, 1742, 1714, 1470, 1062 and 1038 cm-1. This finding is probably consistent with the sub-peaks correspondence in the SD-IR spectrum of LC, GC and CC as shown in Supplementary Table S2. The values of area under the curve (AUC) were calculated with 0.9868, 0.9712, and 0.9853 respectively, shown in Figure 3 (b). Therefore, the PLS-DA model combined with ATR-FTIR data could yielded good sensitivity and specificity when the appropriate AUC threshold was selected as AUC > 95% in this study (CI: 0.9703 ~ 0.997).
To further optimize identification protocol of DTC, and further explore the relationship between subtle changes in serum ATR-FTIR spectra and the detection of early diagnostic markers of DTC, multivariate analysis was performed on four spectral ranges of second-order leads. We carried out the PLS-DA classification with combined data respectively in four spectral ranges, including 3300-2800 cm-1 (mainly caused by lipid hydrocarbon chains), 1700-1600 cm-1 (protein amide I bands), 1500-1400 cm-1 (associated with immunoglobulin IgG) and 1400-1200 cm-1 (from lipids, proteins, nucleic acids and other biomolecules, such as overlapping contribution of glycosaminoglycans, immunoglobulin m). The classification results obtained by the PLS-DA method were shown in Figure 3 (c-f). It can be found that the accuracies of 1700-1600 cm-1 and 1500-1400 cm-1 regions achieved up to 100% (Figure 3 (d-e)), benefited from the abundant subtle spectral changes and DTC-related information in the range of IMF. In addition, the classification accuracy reached 91% for the 3300-2800 cm-1 region (Fig. 3 (c)) and 95% for the 1400-1200 cm-1 region (Fig. 3 (f)). The results demonstrated that the PLS-DA with combined data of IMFs exhibited the potential feasibility to identify different DTCs.”
- In the section 3.4., there is no information about the method of cross-validation and DTC subgroups split.
Response: Thank you so much for your suggestions. We apologized for our unclear presentation and revised the description of the cross-validation in the 2.4. Multivariate and Statistical Analysis as following: ”10-fold cross-validation is used to test the accuracy of the algorithm. The data set is divided into ten parts, and nine of them are used as training data and one as test data in turn. The average of the correctness of the 10 results is used as an estimate of the accuracy of the algorithm.”
Then, we have revised the description of DTC subgroups split information in our revised manuscript as following: “Each group of DTC staging was divided into three different stages, including nonspecific symptom, cancer, postoperative, and control namely1-4 class. Information on the specific subgroups was shown in Supplementary Table S1.” and “Therefore, the in-situ synchrotron-based ATR-FTIR dataset was employed into building 4 types of feature sets (IR absorbance, IR absorbance + shift, SD-IR absorbance, SD-IR absorbance + shift) and respectively using 8 different ML modes for group of DTC staging classifications related to DTCs (Supplementary Table S3).”

Round 2
Reviewer 2 Report
It will be OK almost to publish in this special issue.
It will be better to discuss about the reason of the PLS-DA value sequences of LC > GC > CC > Control.
Reviewer 4 Report
Dear Authors!
Thank you for your efforts to improve the manuscript quality.